# Sinonasal Inverted Papilloma–Associated and De Novo Squamous Cell Carcinoma: A Tale of Two Cities or Not

**DOI:** 10.3390/cancers14215211

**Published:** 2022-10-24

**Authors:** Zekun Wang, Ye Zhang, Jianghu Zhang, Xuesong Chen, Jingbo Wang, Runye Wu, Kai Wang, Yuan Qu, Xiaodong Huang, Jingwei Luo, Li Gao, Guozhen Xu, Shaoyan Liu, Ye-Xiong Li, Junlin Yi

**Affiliations:** 1Department of Radiation Oncology, National Cancer Center/National Clinical Research Center for Cancer/Cancer Hospital, Chinese Academy of Medical Sciences and Peking Union Medical College, Beijing 100021, China; 2Department of Head and Neck Surgical Oncology, National Cancer Center/National Clinical Research Center for Cancer/Cancer Hospital, Chinese Academy of Medical Sciences and Peking Union Medical College, Beijing 100021, China; 3Department of Radiation Oncology, National Cancer Center/National Clinical Research Center for Cancer/Hebei Cancer Hospital, Chinese Academy of Medical Sciences, Langfang 100006, China

**Keywords:** sinonasal squamous cell carcinoma, inverted papilloma, de novo, survival outcome

## Abstract

**Simple Summary:**

Squamous cell carcinoma (SCC) arising either de novo or from benign inverted papilloma (IP) is the most common histological subtype of sinonasal malignancies. Owing to the rarity of the disease, the limited comparative cohort studies provide inconsistent results. Given that none of the studies have well-balanced baseline characteristics and few studies provide adjusted hazard ratios (HRs), it is difficult to draw cogent conclusions. Further studies to identify the difference in prognosis may provide evidence for risk stratification and clinical decision-making. Our study had three main strengths. First, we used the largest patient data set to investigate the differences in clinical characteristics between IP-SCC and DN-SCC. Second, we used propensity score weighting to control for confounders and minimize bias. Third, we compared the annual failure hazards for local failure and distant metastasis between IP-SCC and DN-SCC, and provided a basis for individualized follow-up strategies.

**Abstract:**

Background: Sinonasal squamous cell carcinoma (SNSCC) can arise as either inverted papilloma–associated SCC (IP-SCC) or as de novo SCC (DN-SCC). It is controversial as to whether survival differences between IP-SCC and DN-SCC exist. Methods: Between January 2000 and December 2016, 234 patients with SNSCC were analyzed retrospectively, including 68 with IP-SCC and 166 with DN-SCC. Propensity score matching (PSM) was performed to balance baseline characteristics. The Kaplan–Meier method and Cox proportional hazard model were used to determine risk factors on survival outcomes. Results: The median follow-up time was 98.4 months. Before PSM, lymph node metastasis was noted to be lower in patients with IP-SCC. After PSM, the 5-year DFS, DSS and OS between IP-SCC and DN-SCC were 43.0% vs. 44.5% (*p* = 0.701), 49.2% vs. 56.2% (*p* = 0.753), and 48.2% vs. 52.9% (*p* = 0.978). The annual hazards of local failure, respectively, peaked at 28.4% and 27.8% for IP-SCC and DN-SCC within 12 months after treatment. Afterward, the hazards gradually decreased and the hazard for IP-SCC was always higher before approaching null. Conclusions: This study provides novel evidence to support the clinical utility of improved distinction between IP-SCC and DN-SCC. Further studies are necessary to validate these findings before considering escalation of IP-SCC.

## 1. Introduction

Sinonasal malignancies make up a diverse and rare subset of cancers accounting for 3–5% of head and neck cancers and less than 1% of all malignancies [1,2]. Among all subtypes of the sinonasal tract, squamous cell carcinoma (SCC) is the most common histologic variant, with an average incidence of 0.36 cases per 100,000 patients [3,4]. Sinonasal squamous cell carcinoma (SNSCC) arises as either de novo SCC (DN-SCC) or as inverted papilloma–associated SCC (IP-SCC) [5]. IP-SCC is not simply a mixture of two separate tumors, but a malignant transformation of benign inverted papilloma (IP) [6]. The pathogenesis and etiology of malignant transformation have not yet been established [7]. Previous studies have reported that malignant transformation occurs in 7–17% of IP cases [8,9,10,11]. IP-SCC either frequently occurs synchronously with IP (synchronous IP-SCC) or develops after the previous resection of IP (metachronous IP-SCC) [12]. 

Although IP-SCC seems to be a distinct pathological entity from DN-SCC, contemporary management paradigms suggest that the treatment strategy parallels that of DN-SCC [13,14]. Currently, it is controversial as to whether survival differences between IP-SCC and DN-SCC exist. Owing to the rarity of the disease, only limited comparative cohort studies provide inconsistent results [15,16,17,18,19,20,21]. Given that none of the studies have well-balanced baseline characteristics and few studies provide adjusted hazard ratios (HRs), it is difficult to draw cogent conclusions [15,16,17,18,19,20,21]. Recently, a meta-analysis found an approximately two-fold increased mortality risk in the DN-SCC cohort compared with the IP-SCC cohort [13]. However, insufficiently reported data and the inability to obtain individual patient data precluded our ability to investigate potential differences in tumor characteristics, treatment strategies, and recurrence patterns between IP-SCC and DN-SCC. 

Further studies to identify the differences between IP-SCC and DN-SCC may provide evidence for risk stratification and clinical decision-making. In observational studies, confounding can occur because of differences in the distribution of measured baseline characteristics between groups. Insufficient control for such imbalances leads to biased estimates of effects. Propensity score-matched analysis is widely used in observational studies to minimize the effects of confounding and obtain an unbiased estimate of the effect [22]. Therefore, we designed a propensity score-matched study to investigate whether there are prognostic differences between IP-SCC and DN-SCC using large sample data from our center. To our knowledge, this study is the first propensity score-matched analysis to compare oncologic outcomes between IP-SCC and DN-SCC.

## 2. Materials and Methods

### 2.1. Patient Identification

This study followed the Strengthening the Reporting of Observational Studies in Epidemiology (STROBE) reporting guideline. Between January 2000 and December 2016, medical records of 255 patients with histologically confirmed SNSCC, who were treated with curative intent at our center, were analyzed retrospectively. All patients were restaged according to the American Joint Committee on Cancer 8th Edition (AJCC 8th) staging system. Patients with distant metastasis or second primary malignancies were excluded. Those patients with primary frontal/sphenoid sinus (not applicable to AJCC staging) or carcinoma in situ were also excluded. Finally, 234 patients were recruited for this study (Figure 1). Patients were classified as either IP-SCC or DN-SCC according to medical history and pathologic assessment [18,23]. IP-SCC included both synchronous and metachronous tumors, with metachronous cases defined as those with previous confirmed IP histologically [12]. 

### 2.2. Treatment Planning

The treatment regimens are determined by the advice of the head and neck multidisciplinary team (MDT) and the preference of patients. Overall, 3.8% of patients underwent surgery alone (*n* = 9), 25.6% of patients were treated with radiotherapy (RT, *n* = 60) ± chemotherapy, and 70.5% of patients received surgery combined with radiotherapy (S + RT, *n* = 165) ± chemotherapy. The surgical approach and the extent of resection were dependent on the primary site, extensiveness of the disease, cosmetic considerations, and the discretion of the surgeon. The aim of surgery is to strike a balance between maximized local control and organ preservation. Surgical approaches included open surgery (*n* = 148, 85.1%) and endoscopic surgery (*n* = 25, 14.4%). Tumors extending to the floor of the nose, palate, or maxillary sinus were treated with a maxillectomy and/or lateral rhinotomy. Skull base resection or craniofacial resection was performed in patients with skull base or brain invasion. Orbital exenteration was performed only if primary lesions infiltrated the orbital contents or orbital apex. Typically, patients with cervical lymph node metastasis underwent elective neck dissection along with resection of the primary lesion. 

From 2000 to 2010, two-dimensional conventional radiotherapy (2D CRT) and three-dimensional conformal radiotherapy (3D-CRT) were the mainstays of radiotherapy techniques, while intensity-modulated radiotherapy (IMRT) became mainstream after 2010. The gross tumor volume (GTV) was defined as gross tumor or tumor bed. The high-risk clinical tumor volume (CTV1) was typically defined by adding a 1.0 to 1.5-cm margin to the GTV (2.0-cm margin for IP-SCC tumors receiving definitive RT or incomplete resection), as well as the operative bed at risk of subclinical disease and the lymphatic drainage area of the positive lymph node. An elective CTV (CTV2) was defined as nodal basins of the neck. Elective nodal irradiation (ENI) was routinely delivered to patients with T3–T4 disease and/or cervical lymph node metastasis. Bilateral elective neck irradiation was generally utilized when the tumor infiltrated the midline structure. All patients received standard-fractionated RT (1.82–2.25 Gy per day, 5 days per week) with 70 Gy to the tumor volume, 66 Gy to the tumor bed, 60 Gy to the CTV1, and 50 Gy to the CTV2.

In total, 79 (33.8%) patients were administrated systemic chemotherapy. In our center, neoadjuvant chemotherapy is used for rapidly progressive tumors, defined as tumor growth greater than 30% in the largest dimension within 2 weeks. Concurrent chemotherapy is administered for necrotic primaries, insensitive to radiotherapy, etc. Adjuvant chemotherapy is mainly used in patients with no obvious tumor regression (less than 30%). Single-agent cisplatin was the most commonly used regimen for concurrent chemotherapy; (neo) adjuvant chemotherapy regimens mainly included platinum + paclitaxel, platinum + 5-fluorouracil, and platinum + paclitaxel + 5-fluorouracil.

### 2.3. Statistical Analysis

The primary endpoint was overall survival (OS), defined as the duration from the date of initial treatment to the last follow-up or death from any cause. Other secondary endpoints included disease-specific survival (DSS), and disease-free survival (DFS), defined as the duration from the date of initial treatment to the last follow-up or time of the event (death from disease relapse or progression/disease relapse or progression). Categorical variables were summarized as frequencies (%) and compared using the chi-square test or Fisher’s exact test. Propensity score-matched (PSM) analysis with a caliper of 0.2 was utilized to control for the effect of covariate imbalance and minimize bias between groups. Survival data before or after matching were analyzed with the Kaplan–Meier method and compared with the log-rank test. The Cox proportional hazards model was applied to calculate the adjusted hazard ratio (HR). Smoothed annual hazard curves were plotted by applying kernel-based methods [24]. Statistical analysis was conducted with SPSS Statistics software (version 26.0; IBM Corp., Armonk, NY, USA) and R 4.3.1 (http://www.R-project.org (accessed on 17 May 2022); The R Foundation) software. A two-tailed *p*-value of < 0.05 was considered statistically significant.

## 3. Results

### 3.1. Comparison of Baseline Characteristics

In the whole group, the median age of the entire cohort was 54 years (IQR 46–64 years), and the male-to-female ratio was 2.8:1. The maxillary sinus (59.8%) was the most frequent anatomic subsite, followed by the nasal cavity (29.1%) and the ethmoid sinus (11.1%). Lymph node metastasis at presentation was identified in 21.4% of patients (*n* = 50). The overall stage was I in 3 patients (1.3%), II in 9 patients (3.8%), III in 49 patients (20.9%), IVA in 98 patients (41.9%), and IVB in 75 patients (32.1%). Sixty-eight patients were confirmed as having IP-SCC (29.1%) and 166 patients as having DN-SCC (70.9%). Among the patients with IP-SCC, 53 patients (77.9%) had synchronous tumors, and 15 patients (22.1%) had metachronous tumors.

Baseline characteristics between IP-SCC and DN-SCC groups before and after propensity adjustment are shown in Table 1. Before PSM adjustment, cervical nodal metastasis was noted to be lower in the patients with IP-SCC. In terms of treatment modalities, except S + RT being the most common for both IP-SCC and DN-SCC, more patients with IP-SCC underwent surgery alone, while more patients with DN-SCC received RT. Adjusted for age, gender, primary site, T stage, N stage, TNM stage, year of diagnosis, treatment modalities, and chemotherapy, matched population (1:2 matching ratio) with balanced covariates was generated, including 50 patients with IP-SCC and 90 patients with DN-SCC. The baseline characteristics between both groups were well-balanced.

### 3.2. Survival Outcomes of IP-SCC vs. DN-SCC

With a median follow-up time of 98.4 months (range 1.7–230.1 months), the cumulative rates of DFS, DSS, and OS of the whole group at 5 years were 43.4%, 53.7%, and 51.3%, respectively. Before matching, the 5-year DFS, DSS and OS were 41.8%, 49.3% and 48.5% for IP-SCC, compared with 44.1%, 55.5% and 52.5% for DN-SCC (HR 1.081, 95% confidence interval (CI), 0.738–1.584, *p* = 0.683, Figure 2A; HR 1.119, 95%CI 0.739–1.694, *p* = 0.586, Figure 2B; HR 1.009, 95%CI 0.685–1.486, *p* = 0.963, Figure 2C). Cox models (Appendix A) adjusted for age, gender, primary site, stage, years of diagnosis, treatment modalities and chemotherapy demonstrated IP-SCC was not significantly associated with DFS (adjusted HR 1.207, 95%CI 0.808–1.802), DSS (adjusted HR 1.256, 95%CI 0.815–1.935) or OS (adjusted HR 1.116, 95%CI 0.742–1.678). 

After propensity adjustment, the 5-year DFS, DSS and OS between patients with IP-SCC and DN-SCC were 43.0% vs. 44.5% (HR 1.095, 95%CI 0.684–1.753, *p* = 0.701; Figure 2D), 49.2% vs. 56.2% (HR 1.084, 95%CI 0.650–1.809, *p* = 0.753; Figure 2E), and 48.2% vs. 52.9% (HR 1.007, 95%CI 0.624–1.623, *p* = 0.978; Figure 2F). No significant association of IP-SCC with DFS (adjusted HR 1.362, 95%CI 0.842–2.203), DSS (adjusted HR 1.375, 95%CI 0.813–2.324), and OS (adjusted HR 1.251, 95%CI 0.763–2.051) was observed in PSM-Cox proportional hazards models (Table 2). 

In the S + RT cohort, a sensitivity analysis was conducted to make comparisons between patients with IP-SCC and DN-SCC using the PSM (1:1 matching ratio) method. The matched population comprised 40 patients in each group with balanced covariates (Appendix A). Before PSM, the 5-year DFS, DSS and OS were 50.8%, 56.5% and 55.4% for IP-SCC, compared with 55.9%, 67.5% and 63.3% for DN-SCC (HR 1.230, 95%CI 0.750–2.018, *p* = 0.393; HR 1.338, 95%CI 0.771–2.322, *p* = 0.275; HR 1.151, 95%CI 0.698–1.897, *p* = 0.570). After matching, 5-year DFS, DSS and OS between IP-SCC and DN-SCC were 47.3% vs. 52.8% (HR 1.190, 95%CI 0.630–2.248, *p* = 0.592), 55.2% vs. 67.1% (HR 1.260, 95%CI 0.616–2.581, *p* = 0.526), and 53.9% vs. 60.9% (HR 1.062, 95%CI 0.557–2.025, *p* = 0.854).

### 3.3. Comparison of Failure Patterns

The median follow-up time of the IP-SCC group and the DN-SCC group was 89.9 months and 103.4 months, respectively (*p* = 0.254). The failure patterns of IP-SCC and DN-SCC are depicted in Figure 3. There were no statistical differences in local failure rate (50.0% vs. 41.6%, *p* = 0.238), regional relapse rate (10.3% vs. 7.2%, *p* = 0.436) and distant metastases rate (22.1% vs. 16.3%, *p* = 0.294) between patients with IP-SCC and DN-SCC. The details of patients with regional relapse are depicted in Appendix A. Figure 4 shows the annual hazards of local failure and distant metastases for patients with IP-SCC and DN-SCC. The annual hazards of local failure, respectively, peaked at 28.4% and 27.8% for IP-SCC and DN-SCC within 12 months after treatment. Afterward, the hazards gradually decreased and the hazard of IP-SCC was always higher than that of DN-SCC before approaching null. The annual hazards of distant metastases in IP-SCC patients decreased rapidly from 23.4% to 5.2% within the first 2 years, while the annual hazards of distant metastases in DN-SCC patients decreased slowly and smoothly from 7.9% to 4.9% within the first 2 years. The annual hazards of distant metastases in IP-SCC patients maintained a lower level compared to DN-SCC patients after 2 years of follow-up and onward.

## 4. Discussion

In this propensity score-matched study, we compared the differences in clinical characteristics, survival outcomes, and failure patterns between patients with IP-SCC and DN-SCC. Regional lymph node metastasis was less likely in patients with IP-SCC. IP-SCC had similar DFS, DSS, and OS compared with DN-SCC, irrespective of matching. This was confirmed with Cox multivariate analysis. Further sensitivity analysis for the S + RT cohort yielded consistent results. Although there was no statistical difference in local failure rate between the two groups, the annual hazards indicated that IP-SCC had an aggressive local tendency. These findings provide additional evidence to make an exact distinction between the prognosis of these two pathological types.

In our study, IP-SCC had a relatively low propensity for lymph node metastasis. Lobo et al. also found that lymph node metastasis was more likely in patients with DN-SCC (18% vs. 0%; *p* = 0.02) [16]. Yu et al. reported a lymph node metastasis rate of 13.8% in DN-SCC and 0% in IP-SCC (*p* = 0.073) [18]. However, other studies did not find a significant difference in lymph node metastasis [17,19,20,21]. The conflicting results could be owing to a small sample bias or the fact that patients with carcinoma in situ were not excluded [17,19]. There has also been debate about the difference in staging distribution. Yu et al. and Yasumatsu et al. found a lower proportion of patients with locally advanced (T3–4 or T4) in IP-SCC compared to DN-SCC [18,20]. In contrast, Yan et al. and Quan et al. found a well-balanced staging distribution [17,19]. As the largest patient data set from our center, there was no significant difference in disease stage between IP-SCC and DN-SCC.

Overall, studies based on large public databases reported that the 5-year OS of SNSCC ranged from 30% to 53% [4,25,26,27]. Under the premise of 74% of patients with advanced stages, the 5-year OS rate of our entire cohort was 51.3%, which is comparable to these data. With regard to the prognostic difference between the two pathological types, seven existing comparative studies (Table 3) investigated different survival endpoints between IP-SCC and DN-SCC populations [15,16,17,18,19,20,21]. Five studies only provided results from univariate analyses [16,18,19,20,21]. Among these studies, only one including 86 patients with maxillary sinus SCC showed a significantly better 5-year OS in IP-SCC patients than that in DN-SCC patients [18]. Consistent with the other four studies, imbalanced tumor characteristics ± treatment modalities were also observed in that study [16,18,19,20,21]. Yan et al. retrospectively analyzed 38 patients with IP-SCC and 28 with DN-SCC, and they found that IP-SCC yielded a better 10-year DSS (*p* = 0.041), whereas multivariate analysis demonstrated it did not reach statistical significance [17]. Similarly, another small sample study reported the oncologic outcomes of 27 patients with SNSCC receiving definitive endonasal endoscopic surgery, and multivariable models revealed that IP-SCC was not predictive of DFS or OS [15]. However, it should be noted that the relatively small sample size may affect the stability of the multivariate model in that study [15]. Collectively, the available data cannot make an exact distinction between these two pathological types nor address the debate of whether IP-SCC tumors should be treated as DN-SCC. This study of 68 patients with IP-SCC and 166 patients with DN-SCC comprehensively assessed the clinical outcomes of each pathology, and we found that IP-SCC obtained equivalent DFS, DSS, and OS compared with DN-SCC before and after PSM, which was also confirmed by Cox multivariate analysis. 

Considering the potential impact of surgical approach, margin status, RT technique, and dose, we conducted another PSM analysis in the S + RT cohort to identify the prognostic differences between IP-SCC and DN-SCC, and the results remained consistent. Notably, according to the previous routine practice of our center, the margin from GTV to CTV in IP-SCC patients receiving definitive RT or non-complete resection is larger than that in other patients, mainly due to the local destructiveness of IP itself. Rather, the extent of resection was considered to be adequate for IP-SCC tumors receiving complete resection. In the S + RT cohort, the R0 resection rate of IP-SCC was significantly lower than that of DN-SCC with the balanced staging distribution, which also reflects to some extent that IP-SCC has more extensive local destructive biology. Similarly, in a prior study from Japan, T4 disease occurred less frequently in IP-SCC patients than in DN-SCC patients, but a significantly higher proportion (61%) underwent debulking surgery in comparison to DN-SCC patients (32%) [20]. Thus, the current AJCC system seems to be inadequate for an accurate assessment of the extent of IP-SCC [17,28].

Regardless of IP-SCC or DN-SCC, local failure remains the predominant failure pattern, followed by distant metastasis [19,21]. A prior study including 162 cases of SN-SCC reported the local failure rate was 51.3% for IP-SCC patients, which was significantly higher than for those with DN-SCC (31.7%; *p* = 0.027), with no statistical difference in distant metastasis rates (*p* = 0.261) [19]. Another large sample study found that both pathological types showed similar local recurrence rates (*p* = 0.579), while DN-SCC tumors demonstrated an increased incidence of distant metastasis (19.0% vs. 6.7%, *p* = 0.015) [21]. Although our data indicated that there was no statistical difference in the local failure rate or distant metastasis rate between both pathological types, the annual failure hazards revealed that IP-SCC could have a higher aggressive local failure tendency, especially given that some IP-SCC patients received more extensive irradiation. The risk of distant metastasis for DN-SCC maintained at a higher level compared to IP-SCC patients after 2 years of follow-up. It may be necessary to adjust the intensity of surveillance accordingly based on the comparison of the annual failure hazards between IP-SCC and DN-SCC. A comparison of regional relapse between IP-SCC and DN-SCC was also conducted in our study. Consistent with previous studies, both pathological types had a similar propensity for regional relapse [19,21]. However, the lymph node metastasis rate of IP-SCC at presentation was significantly lower than that of DN-SCC. Whether inadequate neck treatment leads to an increased rate of regional lymph node relapse for IP-SCC is still unknown. According to the details of patients with regional relapse, the proportion of out-field relapse was comparable between the two pathological types. Meanwhile, more than 85% of IP-SCC patients with regional relapse had prior or concomitant local failure events, compared to only 50% in DN-SCC patients with regional relapse. Hence, more aggressive local management might be more urgent than aggressive neck treatment for IP-SCC.

This study does have several limitations. Patients who received radiotherapy were included in this study, which means that the diagnosis of IP-SCC was obtained by the specimens of bite biopsy or partial resection. The presence of IP-SCC may not be known until after the removal of the entire diseased mucosa and after a meticulous pathological examination. Consequently, the detection rate of IP-SCC may be underestimated. In addition, we were unable to control for confounders such as the amount of SCC burden within IP, histologic grade, lymphovascular invasion, and perineural invasion because of retrospective data.

## 5. Conclusions

This study provides novel evidence to support the clinical utility of improved distinction between IP-SCC and DN-SCC. These findings suggest that considering the escalation of IP-SCC (such as the expansion of the margin of clinical tumor volume) might be desirable. Further validation studies are warranted concerning treatment and prognosis implications to translate a tendency as a high-level evidence recommendation.

## Figures and Tables

**Figure 1 cancers-14-05211-f001:**
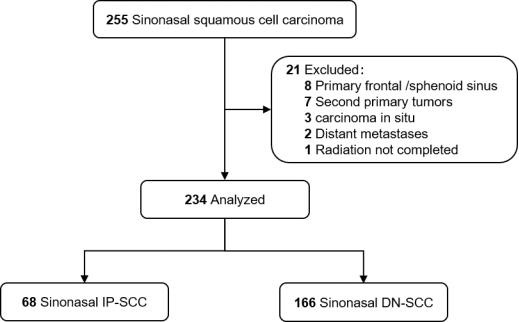
Flow diagram.

**Figure 2 cancers-14-05211-f002:**
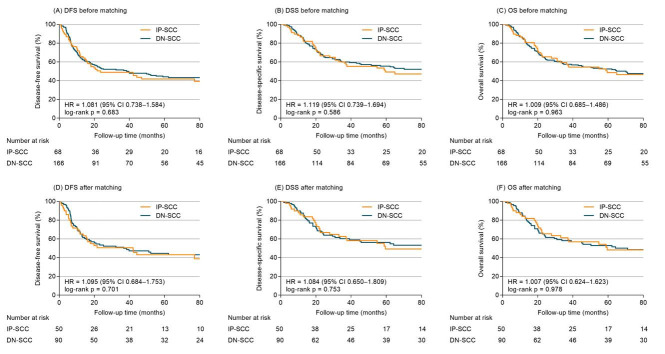
Survival outcomes of IP-SCC vs. DN-SCC for entire cohort before and after PSM.

**Figure 3 cancers-14-05211-f003:**
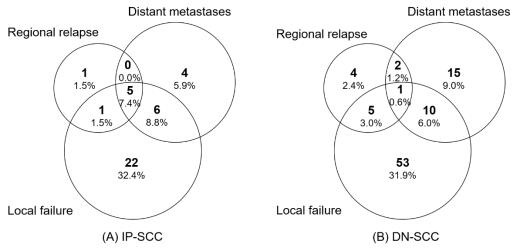
The failure patterns of IP-SCC and DN-SCC.

**Figure 4 cancers-14-05211-f004:**
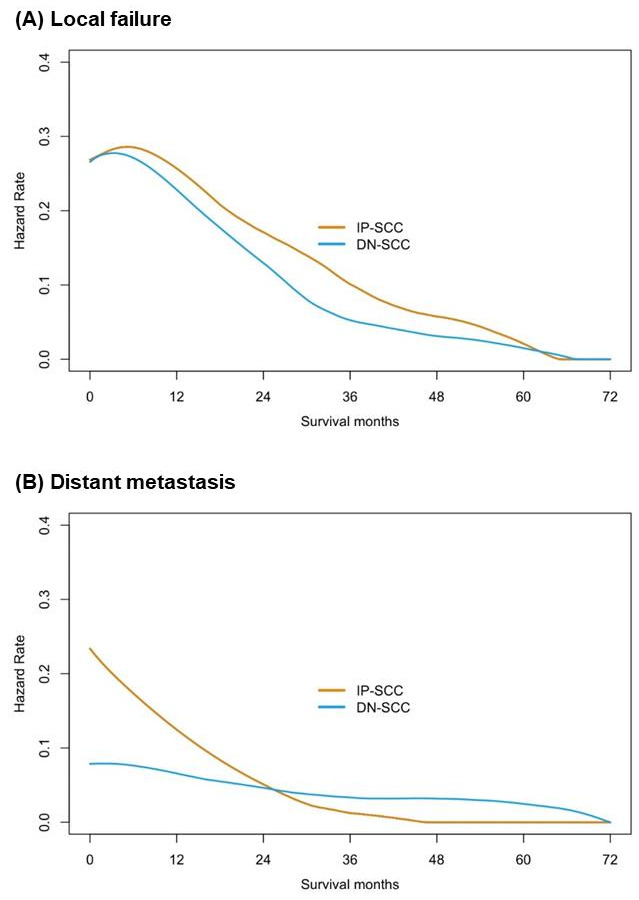
The annual hazards of local failure and distant metastases for patients with IP-SCC and with DN-SCC.

**Table 1 cancers-14-05211-t001:** Comparison of baseline characteristics of inverted papilloma–associated squamous cell carcinoma (IP-SCC) vs. de novo squamous cell carcinoma (DN-SCC) for the entire cohort before and after propensity score-matching (PSM).

Characteristics	Entire Population			PSM Population		
	IP-SCC N = 68 (%)	DN-SCC N = 166 (%)	*p*-Value	IP-SCC N = 50 (%)	DN-SCC N = 90 (%)	*p*-Value
Age, years						
≤54	31 (45.6)	88 (53.0)	0.302	24 (48.0)	38 (42.2)	0.510
>54	37 (54.4)	78 (47.0)		26 (52.0)	52 (57.8)	
Gender						
Male	54 (79.4)	119 (71.7)	0.222	38 (76.0)	72 (80.0)	0.580
Female	14 (20.6)	47 (28.3)		12 (24.0)	18 (20.0)	
Primary site						
Nasal cavity	27 (39.7)	41 (24.7)	0.069	15 (30.0)	30 (33.3)	0.886
Maxillary sinus	34 (50.0)	106 (63.9)		29 (58.0)	51 (56.7)	
Ethmoid sinus	7 (10.3)	19 (11.4)		6 (12.0)	9 (10.0)	
Years of diagnosis						
2000–2009	31 (45.6)	83 (50.0)	0.540	19 (38.0)	40 (44.4)	0.459
2010–2016	37 (54.4)	83 (50.0)		31 (62.0)	50 (55.6)	
T stage						
T1–2	3 (4.4)	13 (7.8)	0.682	2 (4.0)	5 (5.6)	0.991
T3	15 (22.1)	38 (22.9)		10 (20.0)	16 (17.8)	
T4a	30 (44.1)	62 (37.3)		24 (48.0)	44 (48.9)	
T4b	20 (29.4)	53 (31.9)		14 (28.0)	25 (27.8)	
N stage						
N0	61 (89.7)	123 (74.1)	0.008	44 (88.0)	78 (86.7)	0.821
N+	7 (10.3)	43 (25.9)		6 (12.0)	12 (13.3)	
TNM stage (AJCC 8th)						
I–II	3 (4.4)	9 (5.4)	0.847	2 (4.0)	5 (5.6)	0.944
III	13 (19.1)	36 (21.7)		9 (18.0)	14 (15.6)	
IV	52 (76.5)	121 (72.9)		39 (78.0)	71 (78.9)	
Treatment modalities						
Surgery	8 (11.8)	1 (0.6)	<0.001	1 (2.0)	1 (1.1)	0.447
Radiotherapy	7 (10.3)	53 (31.9)		7 (14.0)	19 (21.1)	
Surgery plus radiotherapy	53 (77.9)	112 (67.5)		42 (84.0)	70 (77.8)	
Chemotherapy						
No	56 (82.4)	99 (59.6)	0.001	39 (78.0)	68 (75.6)	0.744
Yes	12 (17.6)	67 (40.4)		11 (22.0)	22 (24.4)	

**Table 2 cancers-14-05211-t002:** Propensity score-matched Cox proportional hazards regression analysis of survival.

	DFS		DSS		OS	
Characteristics	HR (95%CI)	*p*	HR (95%CI)	*p*	HR (95%CI)	*p*
Age, years						
≤54	Ref		Ref		Ref	
>54	1.365 (0.832–2.240)	0.219	2.137 (1.219–3.746)	0.008	2.202 (1.306–3.711)	0.003
Gender						
Male	Ref		Ref		Ref	
Female	0.776 (0.419–1.436)	0.419	0.946 (0.475–1.884)	0.874	1.097 (0.598–2.013)	0.764
Primary site						
Nasal cavity	Ref		Ref		Ref	
Maxillary sinus	1.018 (0.464–2.234)	0.965	1.110 (0.485–2.541)	0.804	1.069 (0.485–2.355)	0.868
Ethmoid sinus	0.885 (0.488–1.604)	0.687	0.779 (0.409–1.484)	0.448	0.843 (0.462–1.536)	0.576
Tumor etiology						
DN-SCC	Ref		Ref		Ref	
IP-SCC	1.362 (0.842–2.203)	0.207	1.375 (0.813–2.324)	0.235	1.251 (0.763–2.051)	0.374
Years of diagnosis						
2000–2009	Ref		Ref		Ref	
2010–2016	0.652 (0.358–1.189)	0.163	0.838 (0.437–1.604)	0.593	0.807 (0.445–1.465)	0.481
TNM stage (AJCC 8^th^)						
I–II	Ref		Ref		Ref	
IV	2.224 (1.090–4.539)	0.028	4.093 (1.607–10.425)	0.003	4.318 (1.812–10.292)	0.001
Treatment modalities						
Surgery + radiotherapy	Ref		Ref		Ref	
Single-modality therapy	2.212 (1.284–3.811)	0.004	2.306 (1.292–4.114)	0.005	2.175 (1.248–3.790)	0.006
Chemotherapy						
No	Ref		Ref		Ref	
Yes	1.749 (0.957–3.197)	0.069	1.678 (0.888–3.169)	0.111	1.636 (0.897–2.985)	0.109

**Table 3 cancers-14-05211-t003:** Summary of studies comparing clinical outcomes of sinonasal IP-SCC and DN-SCC.

Study	Sample Size	IP-SCC%	Years	Follow-Up Time	Imbalanced Baseline Characteristics	5-Year DFS (IP- vs. DN-SCC)	5-Year DSS (IP- vs. DN-SCC)	5-Year OS (IP- vs. DN-SCC)
de Almeida,	27	37.0	2000–2012	Average	Not mentioned	62% vs. 62%, *p* = 0.58	NA	86% vs. 75%, *p* = 0.24
et al.				33.0 m		MVA: ns		MVA: ns
Lobo, et al.	117	24.8	1995–2015	NA	N stage	2.9 vs. 4.8 years, *p* = 0.18	NA	3.4 vs. 5.5 years, *p* = 0.12
Yan, et al.	66	57.6	2000–2015	NA	Surgical technique	10-year: 62.5% vs. 52.6%,	10-year: 89.6% vs. 65.6%,	10-year: 84.6% vs. 62.3%
					Adjuvant therapy	*p* = 0.215	*p* = 0.041, MVA: *p* = 0.061	*p* = 0.065
Yu, et al.	86	24.4	1990–2014	Average	T stage	NA	61.5% vs. 52.8%, *p* = 0.437	58.3% vs. 39.5%, *p* = 0.043
				47.6 m	Treatment modality			
Quan, et al.	162	24.1	2010–2017	Median	Surgery	47.2% vs. 56.9%, *p* = 0.238	NA	58.6% vs. 62.9%, *p* = 0.584
				56.0 m	Chemotherapy			
Yasumatsu,	117	19.7	1990–2016	Average	T stage	NA	3-year: 62.7% vs. 62.0%	NA
et al				44.0 m	Treatment modality		*p* = 0.75	
Li, et al.	173	51.4	2005–2018	Median	Primary site	45.4% vs. 50.1%, *p* = 0.667	NA	63.3% vs. 55.4%, *p* = 0.390
				65.0 m	T2 stage			
					Adjuvant therapy			
This study	234	29.1	2000–2016	Median	Well balanced	43.0% vs. 44.5%, *p* = 0.701	49.2% vs. 56.2%, *p* = 0.753	48.2% vs. 52.9%, *p* = 0.978
				98.4 m		adjusted HR 1.362,	adjusted HR 1.375,	adjusted HR 1.251,
						95%CI 0.842–2.203	95%CI 0.813–2.324	95%CI 0.763–2.051

Abbreviations: MVA, multivariate analysis; ns, not significant; NA, not available.

## Data Availability

All data generated or analyzed during this study are included in this published article.

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
