# Peer review of "Sinonasal Inverted Papilloma–Associated and De Novo Squamous Cell Carcinoma: A Tale of Two Cities or Not"

_cancers, 2022, doi:10.3390/cancers14215211_

Round 1

Reviewer 1 Report

According to my clinician's opinion, the article compares the biology of two tumors (two cities) - IP-SCC and DN-SCC and could be very usful in everyday practice.

I divided my remarks and my conclusions - for and against admission - YES or NOT.

For Yes

1. It is an interesting topic that makes the reader wonder reading the article whether the types of neoplasms actually described are different neoplasms IP -SCC vs de novo SCC (,, Sinonasal squamous cell carcinoma (SNSCC) can arise as either inverted papilloma–associated SCC (IP-SCC) or as de novo SCC (DN-SCC)")

2. Number of analyzed patients - in the compared groups (discussion) the number of patients was significantly smaller 234 vs 86 patients, which indicates greater reliability of the results.

3.

Important conclusions for every clinician - they could have inclinations in clinical practice

The annual hazards indicated that IP-SCC had an aggressive local tendency. These findings provide additional evidence

making an exact distinction between the prognosis of these two pathological types.

,,Although there was no statistical difference in local failure rate between the two groups, the annual hazards indicated that IP-SCC had an aggressive local tendency. These findings provide additional evidence making an exact distinction between the prognosis of these two pathological types." 268-271

327 -334

,, Notably, according to the previous routine practice of our center, the margin from GTV to CTV in IP-SCC patients receiving definitive RT or non-complete resection is larger than that in other patients, mainly due to the local destructiveness of IP itself. Rather, we consider the extent of resection to be adequate for IP-SCC undergoing complete resection and therefore not expand the margin. In the S+RT cohort, the R0 resection rate of IP-SCC was significantly lower than that of DN-SCC with the balanced staging distribution, which also reflects to some extent that IP-SCC has more extensive local destructive biology. ‘’

4. Very importent findings.

279-280

In our study, IP-SCC had a relatively low propensity for lymph node metastasis.

5. Maybe this article is a breakthrough ??

,, To our knowledge, this study is the first propensity score-matched analysis to compare oncologic outcomes between IP-SCC and DN-SCC. ‘’

For NO

1. Inconsistent conclusion - we read in article that:

,,Sinonasal IP-SCC has an aggressive local tendency and lower low pro- pensity for lymph node metastasis, whereas sinonasal IP-SCC had similar survival outcomes compared with sinonasal DN-SCC under modern treatment modalities. These findings suggest that de-escalation of IP-SCC might be not desirable’’

In my opinion

These findings provide additional evidence making an exact distinction between the prognosis of these two pathological types.

Conclusion

,, These findings suggest that de-escalation of IP-SCC might be not desirable.’’

I would rather have written , ,, consider escalating of treatmment (margins, radiotherapy) in IP-SCC’’

2. We could ask - why there are such inaccuracies in results ??

In article - meta-analysis!!!! ,, Survival Outcomes of De Novo vs Inverted Papilloma-Associated Sinonasal Squamous Cell Carcinoma: A Systematic Review and Meta-analysis’’ Jake J Lee et all (This systematic review of 719 patients with dnSCC and 475 patients with IPSCC ) we can read conclusion ,, This systematic review and meta-analysis found that patients with dnSCC had almost a 2-fold increased risk of mortality compared with those with IPSCC. ‘’. This article shows that worse results should be expected in patients with IP – SCC ,, IP-SCC has more extensive local destructive biology ‘’– 334-335

But in this article we can read ‘’sinonasal IP-SCC had similar survival outcomes compared with sinonasal DN-SCC under modern treatment modalities.’’

3. Contesting the results of metaanalysis ,, Survival Outcomes of De Novo vs Inverted Papilloma-Associated Sinonasal Squamous Cell Carcinoma: A Systematic Review and Meta-analysis’’ Jake J Lee

In this article

‘’Recently, a meta-analysis found an approx- imate 2-fold increased mortality risk in the DN-SCC cohort compared with the IP-SCC cohort, and the investigators proposed that de-escalation of IP- SCC treatment may be considered . However, four of the five studies in- cluded in the meta-analysis also suffered from the aforementioned baseline imbalance and lack of adjusted HR , while the remaining one compared metachronous IP-SCC and DN-SCC . Further studies to identify the difference in prognosis between IP-SCC and DN-SCC may provide evidence for risk stratification and clinical decision-making. Acknowledging the limitations of existing studies, we designed a propensity score-matched study to investigate whether there are prognostic differences between IP-SCC and DN-SCC using large sample data from our center. To our knowledge, this study is the first propensity score-matched analysis to compare oncologic outcomes between IP-SCC and DN-SCC. ‘’(74-86)

In conclusion I must say that the article may be useful in clinical practice. It is an innovative approach to the subject IP -SCC vs de novo SCC - approach to the topic of using knowledge from the conclusions obtained for the surgeon (the need to use margins) and for the radiotherapist (margins in contouring). Due to the lack of differences in overall survival in these patients, the reader may get the impression that they are the same neoplasms. On the other hand sinonasal IP-SCC has an aggressive local tendency and therefore it seems necessary to apply more aggressive treatment in this tumor (surgical margins, margins in planning radiotherapy). For me it is a tale of two cities.

Yours faithfully dr hab. n. med Tomasz Rutkowski

Author Response

Dear Reviewer:

Thank you for taking time out of your busy schedule to review the manuscript. Now we have carefully corrected and replied the manuscript for this revision. The revision instructions are as follows:

  1. Inconsistent conclusion

Thanks for your professional and valuable suggestion. After receiving your suggestion, we have carefully considered the accuracy of previous conclusion. After combining your suggestion and the editor's suggestion, we rewrite the conclusion as required.

“This study provides novel evidence to support the clinical utility of im-proved distinction between IP-SCC and DN-SCC. These findings suggest that considering escalation of IP-SCC (such as expansion of margin of clin-ical tumor volume) might be desirable. Further validation studies are warranted concerning treatment and prognosis implications to translate a tendency as a high-level evidence recommendation.”

  1. We could ask - why there are such inaccuracies in results ?

Thanks for your thought-provoking question. First of all, it is not uncommon for different studies to have contradictory results, especially for retrospective studies. It is the controversial conclusions that urges us to further study. As the author explains in the Limitations section, a few studies included in this meta-analysis did not distinguish between invasive carcinoma and carcinoma in situ when reporting survival outcomes. In contrast, our study excluded carcinoma in situ. In our opinions, this may be the main reason for such inaccuracies in results.

  1. Contesting the results of metaanalysis

Thanks for your professional suggestion. We carefully considered the accuracy of our wording. After realizing the inappropriateness, we made the following modifications:

“Recently, a meta-analysis found an approximate 2-fold increased mortality risk in the DN-SCC cohort compared with the IP-SCC cohort. However, insufficiently reported data and inability to obtain individual patient data precluded our ability to investigate potential differences in tumor characteristics, treatment strategies, and recurrence patterns be-tween IP-SCC and DN-SCC”.

Reviewer 2 Report

Dear authors,

The idea of the article is fine and it would be a nice addition to the current literature which is scarce in regards to the topic. In general, the article is a quite long, consequently, it might be hard to follow for the readers. But the content is interesting and solid. Hopefully its length will not dissuade the readers from reading it. Perhaps the authors could shorten it a bit.

Otherwise, there are only some minor flaws in the article that I’d like to address. So, I suggest publishing it after corrections.

1.Introduction

Lines 83-86:                       Propensity score-matched studies (PSMS) are not very common in ENT and HNS literature. Therefore, some more sentences should be written to explain give some more insights to the clinicians that are not experienced in statistics and will have problems understanding the meaning of propensity score-matched studies and, inconsequence, this study. Just to avoid confusion, the following questions should be answered: Why choose the PSMS? Why not do the simple survival analysis from all patients included? What re the benefits of PSMS?

2.Materials and Methods

Lines 90-92:                        The first statement in this section in completely unclear. Please, rephrase.

Lines 143-155:                    In line 143 the authors state that CT was not (routinely) administered. Do they mean “not at all” administered?

·      -   If NO CT at all was administered to the patients, then the following lines from 144 to 155 are completely unnecessary and should be removed from the text. According to the “2.2. Treatment planning” section, I understand it, too, that no CT to the patients was administered. Let’s take a look at the lines 111-113. The authors present the treatment options for all the patients in the study. Namely, 9 patients underwent surgery, 60 RT and 165 S+RT. To sum up, 9+60+165=243. And 243 is the total amount of the patients included in the study. This could also be depicted from Fig.1. No CT options are mentioned in lines 111-113, at all.

·        - If CT WAS administered to the patients, then something should be done to lines 111-113, as the facts from lines 111-113 and line 143 are then in collision.

·        - In either case, I cannot agree with statements in lines 147-149, that concurrent CRT is indicated in the cases of necrosis in the primary tumour, and for insensitive tumours to RT. These are not indications for concurrent CRT. If the concurrent CRT was applied in this settings in the study, it should be properly stated. For example… in our department the CRT is administered for necrotic primaries, insensitive to RT etc.

4. Discussion

Lines 276-277:                    Due to the fact, that patients with frontal and sphenoid primary tumours were excluded, the authors cannot claim that IPSCC were more common in the nasal cavity.

Author Response

Dear Reviewer:

Thank you for taking time out of your busy schedule to review the manuscript. Now we have carefully corrected and replied the manuscript for this revision. The revision instructions are as follows:

1.Introduction

Lines 83-86: Propensity score-matched studies (PSMS) are not very common in ENT and HNS literature. Therefore, some more sentences should be written to explain give some more insights to the clinicians that are not experienced in statistics and will have problems understanding the meaning of propensity score-matched studies and, inconsequence, this study. Just to avoid confusion, the following questions should be answered: Why choose the PSMS? Why not do the simple survival analysis from all patients included? What re the benefits of PSMS?

Thanks for your recognition of our work and your valuable comment. We have modified the introduction and added the following content to the introduction.

“In observational studies, confounding can occur because of differences in the distribution of measured baseline characteristics between groups 23. Insufficient control for such imbalances leads to biased estimates of effects. Propensity score-matched analysis is widely used in observational studies to minimize the effects of confounding and obtain an unbiased estimate of the effect.”

2.Materials and Methods

Lines 90-92: The first statement in this section in completely unclear. Please, rephrase.

Thanks for your careful check. We have corrected these spelling mistakes.

Lines 143-155: In line 143 the authors state that CT was not (routinely) administered. Do they mean “not at all” administered?

  •     -   If NO CT at all was administered to the patients, then the following lines from 144 to 155 are completely unnecessary and should be removed from the text. According to the “2.2. Treatment planning” section, I understand it, too, that no CT to the patients was administered. Let’s take a look at the lines 111-113. The authors present the treatment options for all the patients in the study. Namely, 9 patients underwent surgery, 60 RT and 165 S+RT. To sum up, 9+60+165=243. And 243 is the total amount of the patients included in the study. This could also be depicted from Fig.1. No CT options are mentioned in lines 111-113, at all.
  •       - If CT WAS administered to the patients, then something should be done to lines 111-113, as the facts from lines 111-113 and line 143 are then in collision.
  •       - In either case, I cannot agree with statements in lines 147-149, that concurrent CRT is indicated in the cases of necrosis in the primary tumour, and for insensitive tumours to RT. These are not indications for concurrent CRT. If the concurrent CRT was applied in this settings in the study, it should be properly stated. For example… in our department the CRT is administered for necrotic primaries, insensitive to RT etc.

   Thanks for your valuable suggestion. Our initial description could indeed mislead the readers. In fact, a total of 79 (33.8%) patients were administrated with systemic chemotherapy. Now we modified the description of “2.2. Treatment planning”.

“Overall, 3.8% of patients underwent surgery alone (n = 9), 25.6% of patients were treated with radiotherapy (RT, n = 60) ± chemotherapy, and 70.5% of patients received surgery combined with radiotherapy (S + RT, n = 165) ± chemotherapy.”

   In other hand, we have properly stated the usage of concurrent CRT as required.

  1. Discussion

Lines 276-277:Due to the fact, that patients with frontal and sphenoid primary tumours were excluded, the authors cannot claim that IPSCC were more common in the nasal cavity.

Thanks for your valuable suggestion. In order not to mislead the readers and shorten the length of draft, we have deleted relevant content.

Reviewer 3 Report

This article is interesting, but there are some points to revise before publishing.

1. Authors said that “IP-SCC patients receiving definitive RT or non-complete resection (2.0-cm margin)”. Could you tell readers the reason of larger margin for IP-SCC?

2. Authors said that “A 3 mm isotropic expansion of CTV was used to create the planning target volume (PTV)”. Did you use the 3mm margin in 2D era as same as IMRT era?

3. Could you tell readers the definition of “the high-risk clinical target volume”?

4. Authors wrote some endpoints. What is the primary endpoints?

5. In lines 190, 191, and 208 authors wrote IPSCC and DNSCC.

6. Most of patients in both groups are S+RT. The differences of treatment methods were not statistically in PSM, but the ratio of radiotherapy and S+RT were different, I think. Could authors investigate only about S+RT to reduce the bias? And, most of patients were stage 3 or 4. Could you exclude stage 1?

7. Could you tell readers the differences of follow-up time between both groups?

Author Response

Dear Reviewer:

Thank you for taking time out of your busy schedule to review the manuscript. Now we have carefully corrected and replied the manuscript for this revision. The revision instructions are as follows:

This article is interesting, but there are some points to revise before publishing.

  1. Authors said that “IP-SCC patients receiving definitive RT or non-complete resection (2.0-cm margin)”. Could you tell readers the reason of larger margin for IP-SCC?

Thanks for your recognition of our work and your valuable comment. According to the previous routine practice of our center, the margin from GTV to CTV in IP-SCC patients receiving definitive RT or incomplete resection is larger than that in other patients, mainly due to the local destructiveness of IP itself. Rather, we consider the extent of resection was considered to be adequate for IP-SCC tumors receiving complete resection and therefore not expand the margin. We have already highlighted this point in the section of “Discussion” as required.

  1. Authors said that “A 3 mm isotropic expansion of CTV was used to create the planning target volume (PTV)”. Did you use the 3mm margin in 2D era as same as IMRT era?

Thanks for your valuable question. Conventional therapy was delivered using a wedge-pair or three-field technique in our center. Different from the 3mm margin in IMRT era, the adiation field boundary in the 2D era was determined by reference to the bone markers. Therefore, there is no so-called “margin” in 2D era. In order not to mislead the readers and shorten the length of our manuscript, we have deleted relevant content given that the description of a margin from CTV to PTV is not essential for clarifying the theme of the article.

  1. Could you tell readers the definition of “the high-risk clinical target volume”?

Thanks for your professional suggestion. The high-risk clinical tumor volume (CTV1) was typically defined by adding a 1.0 to 1.5-cm margin to the GTV (2.0-cm margin for IP-SCC tumors receiving definitive RT or incomplete resection), as well as the operative bed at risk for subclinical disease and lymphatic drainage area of positive lymph node. We have added the definition of “the high-risk clinical target volume” to the “2.2. Treatment planning” of the manuscript.

  1. Authors wrote some endpoints. What is the primary endpoints?

Thanks for your valuable suggestion. The primary endpoint was overall survival (OS), defined as the duration from the date of initial treatment to the last follow-up or death from any cause. We have reinterpreted the definition of primary endpoint and secondary endpoint in the section of “2.3. Statistical analysis”.

  1. In lines 190, 191, and 208 authors wrote IPSCC and DNSCC.

Thanks for your careful check. We have corrected these spelling mistakes.

  1. Most of patients in both groups are S+RT. The differences of treatment methods were not statistically in PSM, but the ratio of radiotherapy and S+RT were different, I think. Could authors investigate only about S+RT to reduce the bias? And, most of patients were stage 3 or 4. Could you exclude stage 1?

Thanks for your professional question.

6.1. For your first question, we considered this point when writing the draft. The reason why we did not only investigate the differences between IP-SCC and DN-SCC in the S+RT cohort is that these two pathological types may also accept a single-modality therapy in clinical practice, such as radiotherapy or surgery. To reduce bias, we conducted another propensity matching score analysis in the S+RT cohort, and the results were consistent. This indicates that the number of cases receiving radiotherapy or surgery has little influence on the results.

“In S + RT cohort, a sensitivity analysis was conducted to make com-parisons patients with IP-SCC and DN-SCC using PSM (1:1 matching ratio) method. The matched population comprised 40 patients in each group with balanced covariates (Table S2). Before PSM, the 5-year DFS, DSS and OS were 50.8%, 56.5% and 55.4% for IP-SCC, compared with 55.9%, 67.5% and 63.3% for DN-SCC (HR 1.230, 95%CI 0.750-2.018, p = 0.393; HR 1.338, 95%CI 0.771-2.322, p = 0.275; HR 1.151, 95%CI 0.698-1.897, p = 0.570). After matching, 5-year DFS, DSS and OS between IP-SCC and DN-SCC were 47.3% vs 52.8% (HR 1.190, 95%CI 0.630-2.248, p = 0.592), 55.2% vs 67.1% (HR 1.260, 95%CI 0.616-2.581, p = 0.526), and 53.9% vs 60.9% (HR 1.062, 95%CI 0.557-2.025, p = 0.854).”

6.2. The number of patients with stage 1 and stage 2 is really small, but the reason why we are still included in the analysis is also that these two pathological types also have stage 1 and stage 2 tumors in clinical practice. We want our conclusions to apply to sinonasal IP-SCC and DN-SCC, not to the locally advanced sinonasal IP-SCC and DN-SCC.

Your suggestions are very valuable. From the perspective of data analysis, indeed, only investigating in the S+RT cohort and excluding stage 1 patients can reduce the bias. However, it may deviate from the theme we want to elaborate. To reduce bias, we conducted a sensitivity analysis in the S+RT cohort by using PSM, and combined stage 1-2 or stage 1-3 for analysis.

  1. Could you tell readers the differences of follow-up time between both groups?

Thanks for your valuable question. The median follow-up time of IP-SCC group and DN-SCC group was 89.9 months and 103.4 months, respectively (p=0.254). We have added the above sentence to the “3.3. Comparison of failure patterns” of the manuscript.

Round 2

Reviewer 3 Report

There is no extra comment.